# Defying Death: A Multi-Omics Approach to Understanding Desiccation Tolerance and Senescence in *Eragrostis nindensis*

**DOI:** 10.3390/plants14213360

**Published:** 2025-11-02

**Authors:** Christine F. Madden, Brett Williams, Sagadevan Mundree, Sébastien Acket, Eric Ruelland, Henk W. M. Hilhorst, Jill M. Farrant

**Affiliations:** 1Department of Molecular and Cell Biology, University of Cape Town, Cape Town 7700, South Africa; chrissiemadden@gmail.com (C.F.M.); henk.hilhorst@uct.ac.za (H.W.M.H.); 2School of Biology and the Environment, Queensland University of Technology, Brisbane, QLD 4001, Australia; b.williams@qut.edu.au; 3School of Agriculture and Food Sustainability, University of Queensland, Brisbane, QLD 4072, Australia; s.mundree@uq.edu.au; 4Unité Génie Enzymatique & Cellulaire, Université de Technologie de Compiègne, UMR CNRS 7025, F-60203 Compiègne, France; sebastien.acket@utc.fr (S.A.); eric.ruelland@utc.fr (E.R.)

**Keywords:** *Eragrostis nindensis*, desiccation tolerance, resurrection plants, senescence, RNA processing, lipid droplets, triacylglycerol, RNA sequencing

## Abstract

*Eragrostis nindensis* is a resurrection grass capable of surviving near-complete desiccation. We compared non-senescent leaf tissue (NST) and senescent leaf tissue (ST) to investigate the cellular and molecular basis of desiccation tolerance and senescence. NST recovered fully after drying, while ST failed to regain viability. Integrated transcriptomic (using RNA-Seq), lipidomic (using LC-MS), and ultrastructural (Transmission Electron Microscopical) analyses revealed that NST maintain RNA processing, protein folding, and translational activity during desiccation. Lipidomic data and ultrastructure showed preferential accumulation of polyunsaturated triacylglycerols (TAGs) and lipid droplets in NST, supporting membrane protection and energy buffering. In contrast, ST showed cellular collapse, reduced oleosin protein accumulation, and signatures of senescence. These findings highlight the importance of post-transcriptional and post-translational regulation, as well as lipid metabolism, in preserving cellular integrity during desiccation in this species.

## 1. Introduction

*Eragrostis nindensis* is a polyploid grass capable of surviving near-complete protoplasmic water loss (>90% relative water content (RWC), equivalent to 0.1 g H_2_O/g dry weight) and subsequently restoring full metabolic activity on rehydration. This phenomenon of vegetative desiccation tolerance is underpinned by complex molecular, metabolic, and structural adaptations [1,2,3]. Such plants are colloquially termed resurrection plants, and indeed while all survive drying to 10% RWC, there is a critical threshold of ca. 30% RWC for their survival and at which even drought-adapted species typically experience irreversible damage due to mechanical and oxidative stress [4,5,6,7].

Core desiccation responses across resurrection plants include the upregulation of early light inducible proteins (ELIPS), heat shock proteins (HSPs), late embryogenesis abundant (LEA) proteins, osmoprotectants (particularly sucrose, raffinose family oligosaccharides and trehalose), and various antioxidants [3,8,9,10]. However, species differ in the ratios and specific identities of these protectants, and in how the photosynthetic apparatus is managed during dehydration and recovery. Resurrection plants are broadly classified as either homoiochlorophyllous, retaining chlorophyll and thylakoid structure during drying, or poikilochlorophyllous, such as *E. nindensis*, which degrade chlorophyll and dismantle their thylakoids on drying, reconstituting the photosynthetic machinery on rehydration [11,12].

Earlier work on *E. nindensis* by Vander Willigen et al. [5,13,14] showed that leaf tissues differ in their desiccation response. Older leaves, termed senescent tissue (ST), senesce on desiccation, whereas younger ones do not and are termed non-senescent tissue (NST), retaining viability after drying. Both tissues undergo photosynthetic shutdown, but ST shows earlier membrane instability (electrolyte leakage at 50% RWC) and fails to maintain cellular integrity below 30% RWC [12,13,14]. In contrast, NST maintains cellular integrity through vacuole fragmentation and some degree of reversible wall folding to minimise plasmolysis and cythorrhesis.

Lipid metabolism, although underexplored, has emerged as a key determinant of desiccation tolerance. Lipids are central to membrane biogenesis, cell growth, and signalling [15,16], yet their thermal sensitivity makes them highly responsive to environmental stress [17]. Resurrection plants such as *Xerophyta humilis* [18], *Craterostigma plantagineum* [19], *Boea hygroscopica* [20], and *Paraisometrum mileense* [21] preserve membrane integrity during drying by increasing fatty acid (FA) unsaturation, particularly in thylakoids [2,22,23,24]. However, stress-induced lipid imbalance and reactive oxygen species (ROS) production can damage membranes, as ROS preferentially target unsaturated FAs [25,26]. Consequently, membrane remodelling and lipid-based protection are recurring features of desiccation tolerance [21,27]. A central component of this process is triacylglycerol (TAG), which sequesters toxic intermediates such as free fatty acids (FFAs), thereby mitigating oxidative damage and lipotoxicity [15]. TAG is synthesised in the endoplasmic reticulum and stored in lipid droplets (LDs), organelles stabilised by oleosins that regulate LD structure and size [28,29,30,31,32]. Although well studied in oilseeds, TAG and LDs also accumulate in vegetative tissues under abiotic stress [33,34], where they have been increasingly recognised as dynamic regulators of lipid homeostasis and stress signalling rather than passive storage sites [28,33]

This study integrates transcriptomic, ultrastructural, physiological, and lipidomic analyses to investigate desiccation responses in *E. nindensis*, focusing on the contrasting outcomes in NST and ST. It adds considerably to the knowledge of desiccation tolerance in *E. nindensis* and is the first study comparing multi-omics of non-senescent and senescent tissues of the same resurrection plant. By comparing gene expression, lipid composition, and LD dynamics between these tissues, we aim to uncover mechanisms that enable NST to resist senescence and recover after drying. Special attention is given to RNA and protein regulation pathways, membrane stabilisation, and the role of specific lipid species and LD-associated proteins. These insights may reveal molecular markers of desiccation tolerance and improve understanding of how lipid metabolism and gene regulation underpin desiccation tolerance (and suppression of drought-induced senescence) in this and potentially other resurrection plants.

## 2. Results

### 2.1. Physiological Changes Across a Desiccation–Rehydration Cycle

Initially, RWC remained relatively unchanged, maintaining high leaf water contents (ca. 75% RWC) for eight days after water was withheld (Figure 1A). Rapid water loss subsequently occurred, resulting in an air-dry state by day 13. Both NST and ST dried at similar rates. Plants were kept in an air-dry state for two weeks before being rehydrated by thorough watering (day 28). Only the NST made a full recovery after rehydration whereas the ST did not recover, resulting in leaf death (Figure 1B,D; Appendix A). These findings correlate with previous studies on *E. nindensis* [14,35].

### 2.2. Ultrastructural Changes During Drying and Rehydration

#### 2.2.1. Non-Senescent Tissue

*E. nindensis* is a C4 grass with typical Kranz anatomy, consisting of bundle sheath (BS) cells associated with vascular tissues, surrounded by palisade mesophyll cells [12]. As the BS cells are most informative regarding photosynthesis, they are discussed here. For mesophyll cell observations, see Appendix A (Appendix A). In general, the same trends are seen between the two tissue types, with the obvious exception of starch presence in mesophyll, which experienced considerably more cell wall folding during dehydration. Osmophilic body (OB) accumulation was less pronounced in mesophyll relative to BS cells.

In NST, fully hydrated BS cells had starch-filled chloroplasts arranged centrifugally, with well-formed stacked grana indicating active photosynthesis and starch production (Figure 2A). Upon drying to 60% RWC, the chloroplasts retained their structure and proximity to vascular bundles, though starch depletion was evident (Figure 2B). Vacuolation increased, and small, regularly sized lipid droplets (LDs) began accumulating along the plasma membrane.

At 40% RWC, BS cells were slightly compressed, with chloroplasts loosely congregated near the vascular bundles alongside fragmenting vacuoles. Several large OBs were present between chloroplasts (Figure 2C). Severe drying (25% RWC) caused dramatic changes. Cells became dominated by tightly packed vesicles, and chloroplasts showed complete thylakoid disassembly, characterised by thylakoid vesicles and clustered plastoglobuli. Several LDs accumulated densely along the cell wall and near plasmodesmata, suggesting a role in maintaining cell-to-cell communication, consistent with the role of LDs in intercellular trafficking [28,36]. The larger OBs seen at 40% RWC were not as evident. (Figure 2D).

In the air-dry, desiccated state, chloroplasts remained disassembled (Figure 2E), indicating that ultrastructural reorganisation for desiccation survival occurred by 25% RWC. Despite the critically low water content, the cytoplasm remained well-preserved, showing intact endoplasmic reticulum, plastoglobuli, electron-dense cytosol, and distinct (unfused) vacuoles. Small clusters of dense matter resembled polyribosomes [37]. During rehydration, NST BS cells recovered within 12 h, with chloroplasts reassembling thylakoid membranes and vacuoles coalescing. LDs previously dominating the cell walls disappeared (Figure 2F).

#### 2.2.2. Senescent Tissue

ST BS cells largely resembled NST at full turgor, with starch-filled chloroplasts and distinguishable thylakoid membranes. Several OBs are evident, unlike in the hydrated NST (Figure 3A). At 60% RWC, ST starch appeared to increase in size, giving the appearance of amyloplasts due to the absence of thylakoid membranes (Figure 3B). This is likely due to ongoing photosynthesis; if not occurring in the BS cells, occurring in the mesophyll cells (Appendix A). Indeed, ongoing photosynthesis was recorded at this water content, albeit at a slightly reduced capacity relative to the NST (Appendix A). Mitochondria with darkly stained cristae, indicative of active respiration, clustered around the chloroplasts. Large OBs accumulated in electron-opaque vacuoles, as seen in the NST.

By 25% RWC, ST BS cells exhibited large, irregularly sized OBs and lacked vacuolation or the small LDs seen along NST cell walls (Figure 3C). These ST OBs exhibited features typical of oleosin-deficient droplets, including fusion and enlargement [30]. ST lacked internal structures when compared to NST, with notable subcellular attrition of organelles in the air-dry state. Darkly stained cellular debris—potentially polyphenols—were evident; some wall collapse was evident and the plasmalemma was disrupted, indicating mechanical stress (Figure 3D). We propose that cell death occurs below 25% RWC. The disappearance of starch and most OBs in the desiccated state could indicate transportation to sink tissues, which is typical of nutrient salvage in age-related senescence.

During rehydration, ST cells collapsed further, with no distinguishable organelles or signs of recovery (Figure 3E). Neither BS cells nor mesophyll cells (Appendix A) of the ST survived rehydration. For an overview of LD dynamics during drying and rehydration, see Appendix A.

### 2.3. Transcriptomic Response to Drying and Rehydration

ST was not sampled at 40% RWC, nor at 12 h rehydration. See Section 4.3 for reasoning. Thus, direct comparisons between NST and ST at these sampling points are not made.

#### 2.3.1. Overview of DEG Patterns

A total of 8185 genes were differentially expressed (DEGs) compared to the control (100% RWC, NST), with the majority in NST (7643 genes). Of these, 4953 (60.5%) were shared across both tissue types and all water contents, while 2690 (32.9%) were exclusive to the NST, indicating a more complex regulatory programme for desiccation tolerance (Figure 4A). Here, and at every mention below, genes were classified as differentially expressed when log_2_ fold change exceeded ±2.0, with a false discovery rate (FDR; q ≤ 0.05, [38]) applied relative to the control (100% RWC, NST).

DEGs accumulated or diminished in both tissue types throughout drying and rehydration, increasing as water contents declined (Figure 4B,C). ST had few (*n* = 3) DEGs at 100% RWC when compared to the control, indicating that fully hydrated leaves are transcriptionally alike (Figure 4C). Notably, distinct sets of DEGs emerged during drying, particularly at 25% RWC and at the air-dry state in both tissue types and 12 h post-rehydration in the NST. These findings suggest that *E. nindensis* maintains active transcription even at very low water contents (<25% RWC) and during early rehydration, when RWC has recovered to ca. 20%.

Overall, a greater number of DEGs diminished than accumulated in abundance, consistent with metabolic arrest, which is an established feature of desiccation tolerance [39,40]. The progressive increase in both up- and downregulated DEGs with drying further indicates dynamic transcriptional reprogramming in response to stress.

#### 2.3.2. Core Desiccation Tolerance Mechanisms in NST

In NST, 68.5% of DEGs (2246 genes) were consistently expressed across all RWCs below 60%, reflecting a core suite of drought- and desiccation-tolerance mechanisms (Figure 5, box A). Enriched biological processes included responses to desiccation, ABA signalling, heat stress, solute metabolism and transport, photosynthetic shutdown, reproduction, and metabolite accumulation. Evidence of cell wall macromolecular catabolism signified growth cessation.

Seed-related Gene Ontology (GO) terms were enriched (Figure 5, dashed box), supporting the hypothesis that resurrection plants repurpose seed maturation events for vegetative desiccation tolerance [40,41]. Indeed, seed-related protective genes such as LEAs (other than LEA2), HSPs, dehydrins and seed maturation proteins (SMPs) were also expressed during drying (Appendix A). Metabolic arrest was further supported by shifts towards vacuolation, autophagy, and stress-responsive pathways, including ‘TOR signaling’, ‘vacuolar protein processing’, and ‘protein unfolding’.

During dehydration to ca. 40% RWC, transcripts related to vacuolation and autophagy were enriched (Figure 5, box B), supporting ultrastructural vacuolation observations (Figure 2B). Processes minimising macromolecular denaturation and maintaining protein integrity were dominant, reflecting shifts towards protein protection and metabolic stabilisation. The enrichment of ‘proline biosynthetic process’ genes align with previous findings of proline accumulation in *E. nindensis* [5].

Under severe dehydration (<25% RWC), enrichment of transcriptional regulation (‘mRNA processing’, ‘RNA splicing’, ‘dephosphorylation of RNA polymerase’), solute transport (‘triose phosphate transport’, ‘electron transport chain’), and sugar metabolism (‘glucose-6- phosphate’) suggest preparation for rapid recovery (Figure 5, box C). The storage and protection of RNA-encoding genes must therefore occur below 25% RWC. In the air-dry state, transcript accumulation for DNA replication, protein translation, and cell cycle regulation indicates RNA storage for immediate rehydration-driven modification or translation.

#### 2.3.3. Rehydration and Recovery

In NST, the majority of DEGs remained differentially expressed during rehydration, including 210 DEGs that were exclusively upregulated and 393 DEGs that were uniquely downregulated (Table 1, Figure 5, box A).

Functional categories associated with energy metabolism and detoxification were enriched upon rehydration, such as ‘lactate metabolic process’, ‘molybdenum incorporation’, ‘metal incorporation’, and ‘protein-heme linkage’ (Table 1), despite the absence of photosynthetic carbon. Protein complexes linked to linear electron flow (‘protein-heme linkage’), aldehyde catabolism, and molybdenum cofactor enzymes were upregulated, highlighting oxidative stress mitigation. Enriched GO categories emphasised protein folding, metabolic reactivation, and RNA regulation. Growth-related genes remained suppressed (Table 1), suggesting delayed resumption of cell proliferation post-rehydration. Furthermore, 89 DEGs related to protein synthesis were upregulated during drying, the majority exclusively in NST (56%), of which 13% were only expressed at 12 h rehydration (Appendix A), which may indicate potential efficacy of the translational machinery at low water contents during rehydration in *E. nindensis*.

#### 2.3.4. Genes Suppressed During Dehydration

Fewer GO categories showed significantly downregulated DEGs during drying (Figure 6). These were primarily associated with metabolism, photosynthesis, carbon fixation, and growth, reflecting widespread metabolic suppression (Figure 6, box A). The enrichment of terms such as ‘photosynthesis’, ‘carbon fixation’, ‘growth’, and ‘gluconeogenesis’ indicates a progressive shutdown of energy-intensive processes. This metabolic arrest, including reduced expression of chlorophyll a/b binding and photosynthesis-related transcripts (Appendix A), is characteristic of poikilochlorophyllous species and serves to limit ROS production and conserve energy [42]. Furthermore, reduced abundance of ‘drought recovery’ and cell wall-related genes, including multiple cellulose synthase transcripts, suggest growth inhibition. Suppressed expression of ‘leaf senescence’-associated genes were also more prominent in NST than in ST (Appendix A), aligning with irreversible degeneration in the latter.

Below 40% RWC, transcripts associated with ‘chloroplast relocation’, protein degradation and signalling, such as ‘stomatal complex development’, ‘protein chromophore linkage’, and ‘circadian rhythm’, were downregulated, indicating transcriptional suppression at lower water contents (Figure 6, box B). Categories related to protein and transcriptional upregulation at 40% RWC (Figure 5, box B) showed diminished enrichment at 25% RWC (Figure 6, box C), reflecting a metabolic shift marked by a reduced protein turnover. This coordinated downregulation of protein production genes suggests a protective strategy to maintain protein integrity by limiting molecular crowding, thereby supporting protein stabilisation and detoxification under stress [6,43,44].

Under severe stress (<25% RWC), extensive downregulation of genes involved in translation, signalling, and metabolite transport indicated a shift towards metabolic shutdown (Figure 6, box C). In the air-dry state, the number of genes was too low to yield statistically enriched categories (Figure 6, box D). Nevertheless, the sharp decline in DEGs reflects a widespread cessation of gene expression, marking the transition to a quiescent state.

#### 2.3.5. Exclusive Gene Expression in NST and ST

Approximately 20% of DEGs (excluding 12 h rehydration) were exclusively upregulated in NST, comprising 633 transcripts across 39 enriched GO categories (Figure 7A). Key categories include ‘amino acid biosynthetic process’, ‘response to heat’, ‘chaperone-mediated protein complex assembly’, ‘regulation of seed germination’, and ‘glucose-6-phosohate transport’, all supporting protein synthesis, folding, and RNA stabilisation. The exclusive enrichment of RNA processing and protein metabolism in NST suggests a mechanism for transcript stabilisation and storage during drying.

NST exhibited greater accumulation of DEGs related to post-translational regulation compared to the ST (Figure 5, boxes B, C). These genes are associated with signalling, protein degradation, post-translational modification, and transport. Furthermore, enriched categories such as ‘RNA processing’, ‘protein folding’, and ‘chaperone-mediated protein complex assembly’ were prominent at lower water contents (Figure 7B), suggesting active mechanisms for maintaining protein integrity. This over-representation highlights the importance of transcriptional regulation (RNA processing) and protein metabolism (translational machinery transcripts) in desiccation tolerance.

Of the 93 DEGs belonging within the enriched ‘RNA processing’ and related categories, 28 were associated with transcriptional regulation, including several stress-induced transcription factors (e.g., AP2, bZIP) and genes involved in post-translational modification, such as protein synthesis and degradation (Appendix A). Notably, the small nuclear ribonucleoprotein family protein (snRNP; AT3G14080), exhibited the highest fold change in NST during desiccation; 11-fold at 40% RWC and 10.7-fold at 25% RWC.

### 2.4. Lipidomic Shifts During Drying and Rehydration

Lipids were extracted from NST and ST samples, at 100% RWC, 25% RWC, AD, 12 h and 24 h rehydration. These were separated by reverse phase chromatography and analysed by mass spectrometry. A total of 67 lipids were annotated; TAG, DAG, MGDG, and DGDG were obtained in the positive mode, while SQDG, PE, PC, PI, and PG were obtained in the negative mode. A heatmap was generated (Figure 8), with clustering of the metabolites. In NST, drying is characterised by a relative decrease between 100% RWC and AD in MGDG 18:1/18:3, MGDG 18:3/18:3, SQDG 18:3/18:3. This is paralleled with the relative increase in PE (18:2/18:3, 18:1/18:3, 18:1/18:2), PC (16:0/18:2, 18:2/18:3, 18:1/18:3, 18:1/18:2 and 18:1/18:1), and TAG (16:0/18:2/18:3, 20:0/18:1/18:3, 16:0/18:1/18:3, 16:0/18:1/18:2, 18:1/18:1/18:2 and 18:1/18:2/18:2). For NST samples, 12 h rehydration leads to an increase in TAG (18:1/18:1/18:1, 20:1/18:3/18:3, 24:0/18:1/18:3, 20:0/18:1/18:2, 22:0/18:1/18:2). The profile after 24 h rehydration differs from that at 12 h rehydration, and from that in NST at 100% RWC, suggesting at 24 h rehydration is not sufficient to return to the 100% RWC profile. As for ST samples, dehydration is characterised by the marked decrease in DGDG (18:1/18:3, 18:2/18:3, 18:3/18:3), MGDG (18:1/18:3, 18:2/18:3, 18:3/18:3) and increase in some TAGs (18:2/18:3/18:3, 18:3/18:3/18:3 and 20:0/18:3/18:3). This clearly indicates differences between NST and ST during dehydration. Furthermore, changes during rehydration of ST are minimal. This is confirmed by the PCA (Figure 9). For legibility, we separated NST samples (Figure 9A) and ST samples (Figure 9B), however, the positions are directly comparable across panels. PC1 explains 28.7% of the variance while PC2 explains 18.2% thereof. For NST samples, drying leads to a change towards more positive values of the PC1 axis. This is also the case for ST samples; however, 25% RWC is similar to that of the AD tissues, which is not true for the NST samples. Indeed, for ST, with exception of the 100% RWC tissue, all samples are comparable, suggesting that dehydration led to irreversible changes in lipid profiles.

### 2.5. Protein Expression as a Proxy for Translational Regulation in NST

#### Oleosin Protein Expression

Of the eleven oleosin transcripts identified in the transcriptome, eight were differentially expressed in the NST and six in the ST during drying and rehydration (Figure 10A). The OLE1 antibody used for immunoblotting showed cross-reactivity with three *E. nindensis* isoforms (Eni_009916, Eni_003038, and Eni_007625), which share 57–67% sequence similarity with the *Arabidopsis* OLE1 protein used to generate the antibody.

Immunoblotting confirmed the accumulation of oleosin proteins during drying, predominantly in the NST (Figure 10A). Protein expression became detectable at 40% RWC, peaked under severe dehydration and desiccation, and generally declined during rehydration (Figure 10B). In contrast, ST samples exhibited only faint signals at the same band positions, despite comparable transcript abundances.

Consistent protein bands were detected at ca. 33 kilodaltons (kDa) across biological replicates, notably larger than the expected 19 kDa of OLE1. This may indicate binding to other oleosin isoforms, as oleosins typically range from ca. 15–30 kDa [45], or may result from mono-ubiquitination, post-translational modification, or interaction with oleosin-like proteins such as caleosin, as observed by Yu et al. [46]. Despite multiple bands, the consistent detection of the dominant band supports translational control and differential protein accumulation during drying.

## 3. Discussion

### 3.1. Cellular Ultrastructure

This study demonstrated how *E. nindensis* undergoes marked physiological and ultrastructural modifications to withstand desiccation-induced oxidative and mechanical stress, as has been reported in other resurrection plants [1,6,42,47]. These responses involve genetic reprogramming, notably photosynthetic shutdown and senescence inhibition [7,43,48,49]. While NST and ST exhibit similar ultrastructure when hydrated (Figure 2 and Figure 3), distinct differences emerge during drying, reflecting divergent metabolic regulation.

The similar drying rates observed in NST and ST (Figure 1), indicate that senescence in ST is not triggered by differential water loss. Subcellular differences become pronounced under moderate stress (<40% RWC; Figure 2C), identifying this as a pivotal transition phase. At this point, mechanisms for biophysical protection, metabolic reorganisation, and preparation for rehydration are activated, and these are hallmarks of desiccation tolerance [3]. LDs appear in NST below 60% RWC, resembling those seen in seeds during maturation or environmental stress [36], and accumulate along the cell periphery, potentially stabilising the plasmalemma and membrane integrity [28,50]. Supporting this, BS cells in NST exhibit minimal membrane retraction, implying mechanical stabilisation. In contrast, the absence of LDs in ST likely contributes to the irreversible collapse of membrane and organelle integrity.

### 3.2. Desiccation Tolerance Signature

As a truly desiccation-tolerant species, *E. nindensis* shows distinct metabolic shifts across RWCs. Metabolic arrest and the accumulation of protective molecules during drying are consistent with patterns observed in other resurrection plant species [51]. At 60% RWC, photosynthesis and growth cease (Appendix A), while protective pathways, such as unfolded protein responses, are activated at 40% RWC. Further drying (<25% RWC) enhances transcriptional control and, in the desiccated state, DNA repair pathways are enriched. Notably, NST-specific transcripts are enriched for genes related to RNA processing and translation (Figure 4). This implies that NST maintains translational control during drying, though further evidence is required to support this.

The dominant GO terms enriched at 12 h post-rehydration were ‘response to gravity’ and ‘protein-heme linkage’ (Table 1). Heme-containing proteins are essential for redox reactions, electron transport, oxygen binding, signal transduction, and micro-RNA processing [52]. However, unbound heme can trigger iron-dependent lipid peroxidation, leading to ferroptosis and cell death [52,53]. Given the abundance of polyunsaturated lipids in *E. nindensis*, particularly during drying, the upregulation of ‘protein-heme linkage’ may reflect protective regulation against lipid peroxidation and ROS generation during rehydration. In addition, the reduction in ‘cellulose catabolism’ suggests decreased cell wall remodelling, potentially limiting water loss [54].

### 3.3. Seed-Related Signature

Seed-related GO terms, central to dormancy and desiccation tolerance, are broadly enriched in NST across all RWCs (Figure 4) [40,55]. However, 60.5% of DEGs are shared between NST and ST (Figure 4 and Appendix A), indicating that senescence and cell death in ST are not caused by a failure to transcribe genes that enable desiccation tolerance. Rather, they likely result from an inability to maintain effective RNA processing (i.e., transcription and post-transcriptional regulation) or translational capacity. We propose that the preservation of these processes, beyond transcriptional activation alone, is essential for desiccation tolerance.

To demonstrate this, both HSPs and LEAs exhibit similar expression patterns in NST and ST (Appendix A), supporting their known chaperone roles during desiccation [56,57,58,59,60,61,62,63]. Similar expression profiles in the desiccation-sensitive *Eragrostis tef* further imply that transcript abundance alone does not confer tolerance [64].

### 3.4. Translational Regulation and Senescence

The overrepresented suppression of ‘leaf senescence’ in NST indicates active regulation of senescence (Figure 6 and Appendix A). In addition, the enrichment of post-transcriptional and translational machinery in NST suggests active protein synthesis is maintained during desiccation, a characteristic of tolerant species [65,66,67,68]. Upregulated ribosomal proteins, elongation factors, and chaperones, coupled with the appearance and organisation of cellular organelles (Figure 2), suggest that NST sustains RNA processing at low RWCs. In contrast, ST exhibits reduced RNA regulation, increased RNA degradation, and protein turnover (ubiquitination) pathways indicative of senescence (Appendix A) [48,69].

Transcript accumulation alone does not equate to functional tolerance [70]. In *Tortula ruralis*, transcripts accumulated during drying for the resumption of metabolism upon rehydration [65]. Transcripts must therefore be stabilised during desiccation, likely in ribonucleoprotein complexes [44,45,46]. Similar transcriptional trends observed in *E. nindensis* (Figure 4 and Appendix A) may allow rapid mobilisation upon rehydration [71,72,73], potentially through RNA sequestration in processing bodies (P-bodies), as observed in seeds [70,71,72,73]. The role of P-bodies in RNA storage under stress warrants further investigation.

RNA degradation also shapes transcript pools [74], yet mechanisms determining RNA storage, degradation or translation during drying remain unresolved [75]. Stabilising and mobilising RNA during- and post-desiccation is essential for desiccation tolerance [76,77]. In *X. humilis* PSII-related RNAs persist through desiccation, enabling recovery after rehydration through, inter alia, facilitating protein synthesis post-rehydration [78,79]. Evidence of dense polyribosomes near the endoplasmic reticulum (Figure 2E), and enriched GO terms for chaperone activity (Figure 5, box B), further supports ongoing translation and protein repair.

The exclusively upregulated snRNP (AT3G14080) in NST showed the highest fold changes among RNA processing genes (11 and 10.7 at 40% and 25% RWC, respectively; Figure 5 andAppendix A). As part of the LSM1-7 decapping activator complex, it regulates RNA turnover and abiotic stress responses in *Arabidopsis* [80,81], underscoring the importance of RNA regulation in *E. nindensis* desiccation tolerance. Furthermore, elevated transcript accumulation in NST (Figure 4A) coupled with its survival post-desiccation, suggests active transcript storage during drying.

### 3.5. Energy Metabolism and Rehydration

TEM revealed stark differences in LD accumulation between NST and ST (Figure 2 and Figure 3 and Appendix A). In NST, starch was present under fully hydrated conditions, persisted during drying, and started to be metabolised (indicated by the white halo) but was still present within 12 h of rehydration. As photosynthesis does not resume this early, the starch must have been pre-stored. LDs had a different trend; they began accumulating from 60% RWC, increasing throughout desiccation. Their rapid disappearance post-rehydration suggests that stored lipids are mobilised to support membrane repair during early recovery. Recent studies have shown that LDs rapidly disappear following the cessation of stress [82,83,84], supporting the hypothesis that LD degradation contributes to recovery during the return to favourable conditions [28]. Transcriptomic data support this observation: starch metabolism declined during drying (Figure 6, box A), while lipid storage increased significantly (Figure 5, box A and Figure 8), indicating a metabolic shift from carbohydrate to lipid-based energy reserves.

Starch and lipid biosynthesis are competing pathways, with starch synthesis typically prevailing unless carbon supply exceeds utilisation [69]. The presence of large OBs among starch-containing chloroplasts at 40% RWC in both tissues may reflect the accumulation of excess fatty acids from thylakoid membrane breakdown. In *E. nindensis*, carbon allocation towards lipid reserves below 40% RWC likely represents an adaptive strategy for energy storage and homeostasis under extreme stress. This reliance on lipid metabolism, coupled with seed-related pathways, underscores the essential role of lipids in energy management and cellular protection during desiccation.

### 3.6. Desiccation-Induced Lipid Droplet Formation and Ultrastructural Remodelling

Across all NST conditions, the lipid profile remains dynamic, which is consistent with that of living, adaptable tissues. We observe a decrease in plastid lipids, notably MGDG and SQDG, together with a concomitant increase in TAG and some phospholipids, and this pattern is compatible with plastid membrane remodelling and diversion of acyl chains into OBs. TAGs, which are typically associated with developmental contexts (seeds; senescing leaves) or stress-induced regulation [83,85,86,87], increase under dehydrated conditions.

Plastid lipids also decline in the ST, but with a distinct signature. DGDG is more impacted by drying in ST than in NST.. TAG increases in ST, but with a different molecular species distribution than in NST. Strikingly, little additional change is observed across 25% RWC, air-dry, and rehydration, indicating that once tissues pass a severe dehydration threshold (ca. 25% RWC), the lipid state becomes largely static and persists through the air-dry state and early rehydration.

Ultrastructural observations support these findings. Uniform LDs congregating along the plasmalemma in NST at 40% and 25% RWCs contrasted with large, fused OBs in ST (Figure 2 and Figure 3), likely resulting from the absence of oleosin expression (Figure 10). Oleosin is known to regulate LD formation and size, and its protein expression reflects translational activity at low RWCs [30]. ST failed to express oleosin adequately despite showing similar transcription trends to NST. This may imply that ST is incapable of post-transcriptional and post-translational modification, resulting in cellular disintegration.

Sequestering potentially toxic FFAs into LDs appears to enable desiccation tolerance in *E. nindensis*. LDs also provide critical mechanical stabilisation and energy-providing roles in NST. In contrast, ST cells fail to perform these protective functions and die. Together, these results indicate that desiccation tolerance in *E. nindensis* relies on preserving transcriptional and translational capacity in NST to maintain key processes such as protein expression, lipid metabolism, and appropriate metabolite compositions. Without this, stress-responsive genes seem to fail to be post-transcriptionally modified or translated into protective proteins in ST, leading to senescence.

In a recent metabolomics study on *E. nindensis,* it was shown that there were differentially abundant metabolites in NST, such as sucrose, glutamic acid, aspartic acid, proline, alpha-ketoglutaric acid, and allantoin, which are major drivers of desiccation tolerance, and furthermore aid the plant in recovering during rehydration. In contrast, metabolites which accumulated in ST—leucine, DL-isoleucine, L-valine, and cinnamic acid—indicated initiation of programmed cell death (PCD), leading to senescence [44]. However, these molecular players act downstream from the actual induction site of senescence by the environment; the upstream signalling behaviour leading up to it is still largely unknown. It is clear, however, that resurrection plants successfully suppress drought-induced senescence, which is a unique feature to maintain desiccation tolerance and successfully continue growth and development after rehydration. However, to accommodate this, there is a requirement for nutrients from senescing tissue. Senescence can be induced in leaves that are senescence-competent, i.e., those at more advanced stages of ageing. It is not yet clear how the plant regulates leaf-age timing but transcriptional networks, including those for stress-induced senescence, have been proposed [88]. It remains to be seen if the genes of these networks are also involved in the induction or repression of senescence in (drying) resurrection plants. Ultimately, such an understanding could facilitate senescence suppression in cereal crops under drought conditions.

## 4. Materials and Methods

### 4.1. Germination and Propagation

Seeds of *E. nindensis* were collected, with permission from the land manager, Mr P. Venter, from natural populations in Aggeneys, Northern Cape, South Africa (29°16′41.1″ S 19°00′25.4″ E). Surface-sterilised seeds were germinated on 4.4 g/L MS medium solidified with 0.8% agar in a growth chamber set at 24/15 °C day/night with a 16 h photoperiod (light intensity: 300 μmol m^−2^ s^−1^) for 2–3 weeks until seedlings developed. After acclimation by removing lids, seedlings were transferred to trays with soil containing vermiculite and 0.4% Phostrogen All Purpose Plant Food (Bayer, Midrand, South Africa) supplemented with 2% calcium. Plants were watered three times a week, with a weekly light spray of Phostrogen (N16, P10, K24) nutrient solution. After one month, seedlings were transplanted into 10.5 cm pots containing a sand–vermiculite–perlite mixture (2:1:1) and grown for six months to accumulate sufficient biomass for the dehydration experiment. Two weeks before experimentation, plants were transferred into a Percival Intellus Control System (Model I-41LL, Percival Scientific, Inc., Perry, IA, USA) with a 16 h photoperiod using a light intensity of approximately 300 μmol.m-2.sec-1, with 24 °C day-time and 15 °C night-time temperatures for acclimation purposes, and they were kept well-hydrated under these controlled conditions. Plant positions were randomised throughout acclimation and experimentation.

### 4.2. Determination of Leaf Phenotype

Leaf phenotype was defined by position and maturity (Appendix A). NST comprised young, fully mature (unrolled) leaves positioned highest on the tiller, while ST comprised the lowest (oldest) leaves. To ensure leaf age uniformity, plants underwent a prior desiccation and rehydration cycle six weeks before the experiment. All dead leaves were pruned prior to stress treatment initiation to synchronise ST age and minimise age-dependent senescence. Senescent leaf tips were pruned. All subsequent physiological, ultrastructural, lipidomic, and transcriptomic analyses were conducted on these plants.

### 4.3. Desiccation and Rehydration Treatment

Desiccation was induced by withholding water until the leaves reached an air-dry desiccated state (RWC < 10%, AD). Tissues were collected when fully hydrated (100% RWC), mild (60% RWC), moderate (40% RWC), severe (<25% RWC) dehydration, desiccation (air-dry, <10%, AD) and 12 h post-rehydration (12 h) after thoroughly watering the plants. Due to financial restrictions, two sampling points had to be eliminated for the transcriptomic studies. We thus chose not to sample ST leaves at 40% RWC, which is just above the more critical threshold of below 30% RWC, and ST 12 h rehydration, as these tissues are effectively dead [12]. We also extended the period of rehydration to 24 h for the lipidomics study.

A minimum of three individual plants were used as biological replicates per RWC category. From each plant, approximately three leaves were collected from both NST and ST tissue types. Leaves were immediately weighed (fresh weight, accurate to 10 µg), then placed in microtubes filled with water and kept at 4 °C in the dark overnight. After blotting to remove surface moisture, the turgid weight was recorded. Samples were then sealed in labelled foil packets, oven-dried at 70 °C for 48 h and weighed to obtain dry weight. RWC was calculated gravimetrically according to Barr and Weatherley [89]:RWC (%) = ((fresh weight − dry weight)/(turgor weight − dry weight)) × 100(1)

Plants were left air-dry for two weeks before rehydration. Individual plants with leaf discoloration, stress symptoms, or significantly different biomass were excluded. Sampling occurred at one-hour post-artificial dawn at each RWC stage, guided by physiological changes like leaf curling and chlorophyll loss [35]. Tissue samples from leaves with similar RWCs (±10%) were pooled into discrete RWC bins, with a minimum of three replicates per stage. Samples were either snap-frozen in liquid nitrogen and stored at −80 °C or preserved for microscopy. A two-way ANOVA was conducted to determine the difference between RWC per tissue type over time.

### 4.4. Transmission Electron Microscopy

Leaf segments (~4 mm^2^) from three plants at each RWC stage underwent fixation and processing following Cooper and Farrant [90], with slight modifications. Tissues were fixed in 2.5% glutaraldehyde in 0.1 M phosphate buffer (pH 7.4) containing 0.5% caffeine, post-fixed in 1% osmium tetroxide, and dehydrated in an ethanol gradient. Samples were sequentially immersed overnight in 50%, 75%, and 87.5% Spurr’s resin solutions (diluted with acetone) at 4 °C, then embedded in 100% Spurr’s resin. Resin blocks were polymerized at 60 °C overnight, with the adaxial leaf surface oriented upwards. Cross-sections were cut using a Reichert Ultracut-S microtome (Leica Microsystems, Wetzlar, Germany), stained with 2% uranyl acetate for 10 min, followed by 1% lead citrate [91]. Ultrastructural changes were examined using an FEI Tecnai T20 transmission electron microscope (FEI Company, Hillsboro, OR, USA).

### 4.5. RNA Extraction and Transcriptomics

Total RNA was extracted from 50 to 100 mg of leaf tissue using TRI Reagent^®^ (Sigma-Aldrich, Merck, Cat. No. T9424, St. Louis, MO, USA) per the manufacturer’s protocol. Three replicates were used. RNA quality was assessed via Q-bit Flurometer (Thermo Fisher Scientific, Waltham, MA, USA) and Agilent 2100 Bioanalyzer (Agilent Technologies, Santa Clara, CA, USA). Libraries for RNA sequencing and RT-qPCR were prepared with barcoding for paired-end (150 nucleotide) and single-end (75 nucleotide) reads, sequenced on an Illumina HiSeq 2000 platform (Illumina, San Diego, CA, USA). Read quality was assessed using FASTQC v0.11.5, and adapter trimming was performed with Trimmomatic v0.36 [92].

Reads were aligned to the *E. nindensis* reference genome [64] using HISAT2 v2.1.0, and transcript assembly and quantification were conducted with StringTie v1.3.3 as part of the Tuxedo pipeline [93]. EdgeR normalised gene counts, and transcript abundances were log_2_-transformed [94]. Genes were classified as differentially expressed when log_2_ fold change exceeded ±2.0, with a false discovery rate (FDR; q ≤ 0.05, [38]) applied relative to the control (100% RWC, NST).

### 4.6. Annotation and Quality Control

To maximise annotation confidence, multiple tools were employed. InterProScan v5 [95] identified conserved domains, Swissprot, GO, and BLAST hits, using an e-value threshold <1 × 10^−10^. Sequences were also aligned to *Arabidopsis thaliana* using BLAST+ v2.6.0 to identify homologs, with annotations assigned when more than ten *E. nindensis* reads mapped to the same *Arabidopsis* gene. Mercator v4.0 [96] was used to generate functional annotations and GO terms to analyse gene functions and biological processes [97,98]. Assembly completeness was evaluated with BUSCO v3.0 using the Embryophyta database [99].

### 4.7. Functional Enrichment Analysis of Co-Expressed Genes

DEGs that were common or exclusive to NST and ST were identified using Venny (v2.1; [100]). Heatmaps were generated in Genesis (v1.8.1; [101]) to visualise expression profile similarities among DEGs. Only NST rehydration (12 h) was sequenced due to financial constraints and the absence of viable tissue in ST. Consequently, NST rehydration was analysed independently.

GO enrichment analysis was performed using the Biological Networks Gene Ontology (BiNGO) tool in Cytoscape (v3.5.1) to visualise overrepresented GO categories in each sample [102]. BiNGO identifies statistically enriched GO terms based on the transcriptome-wide GO composition and applies hypergeometric tests with false discovery rate (FDR) correction (<0.05) for multiple testing.

To reduce redundancy arising from the hierarchical nature of GO terms, REViGO [103] was used to cluster semantically similar terms and eliminate redundant categories. The degree of over-representation was quantified as the ratio of observed to expected gene counts. A similarity index of 0.7 was applied, and redundant categories were filtered using a dispensability threshold of *p* < 0.05 unless otherwise specified.

### 4.8. Lipidomic Profiling and Analysis

Lipids were isolated using the standard Bligh and Dyer procedure [104] with a solvent mixture of methanol, chloroform, and water (2:1:0.8, *v*/*v*/*v*). This and subsequent analyses were repeated thrice. After phase separation, the organic layer was collected, evaporated under a gentle stream of nitrogen, and stored at −20 °C until further use. Prior to analysis, dried extracts were resuspended in 100 µL of isopropanol. Lipidomic analysis was performed according to [105]. Briefly, chromatographic separation was performed on a C18 Waters Acquity analytical column (100 × 2.4 mm, 1.7 µm) and a CSH C18 VanGuard precolumn (5 × 2.1 mm, 1.7 µm), maintained at 65 °C. The mobile phase consisted of solvent A (60:40 acetonitrile/water, *v*/*v*) and solvent B (90:10 isopropanol/acetonitrile, *v*/*v*). For analyses in positive-ion mode, solvents were supplemented with 10 mM ammonium formate and 0.1% formic acid, whereas in negative-ion mode they contained 10 mM ammonium acetate and 0.1% acetic acid. A constant flow of 0.6 mL/min was applied, and 10 µL of sample were injected for each run. The LC system was coupled to a quadrupole time-of-flight (QTOF) mass spectrometer (Agilent 6538, Agilent Technologies, Santa Clara, CA, USA) equipped with dual electrospray ionisation. The source was operated at 350 °C with a fragmentor voltage of 150 V and a skimmer voltage of 65 V. Data were acquired in full-scan mode across an *m*/*z* range of 100–1700 at a rate of two spectra per second. The most abundant precursor ions were selected for fragmentation at 35 eV, and MS/MS scans were performed on the six most intense ions. Continuous mass calibration was achieved using two reference ions (*m*/*z* 121.0509 and 922.0098 in positive mode; *m*/*z* 112.9856 and 1033.9881 in negative mode). Instrument parameters, data acquisition, and initial processing were controlled with MassHunter B.07 software (Agilent Technologies). Raw data files were converted from the native *.d format to *.mzML using MSConvert and processed with MS-DIAL (v5.5) [106]. Data preprocessing steps included baseline correction, peak detection, alignment, gap filling, and adduct assignment. Lipid annotation was based on accurate mass, retention time, and MS/MS spectra comparison. Potential duplicates, ghost peaks were filtered by the MSCleanR workflow implemented in MS-DIAL. Data were filtered based on a minimum blank ratio of 0.8, removed ghost peaks and inaccurate masses, and applied a maximum deviation threshold of 50. Curated lipidomics data from both positive (+) and negative (–) ionisation modes were combined into a single dataset. For Principal Component Analysis (PCA), the data were standardised using the *StandardScaler* module from *scikit-learn*, which centres each feature (metabolite) to a mean of 0 and scales it to unit variance (standard deviation of 1). The standardised data were then subjected to PCA to reduce dimensionality and visualise sample clustering. The first two principal components were plotted, with 95% confidence ellipses drawn around each condition group. Additionally, normalised (Z-scored) data were visualised as a heatmap using *seaborn*, with the order of conditions preserved as specified. The nomenclature used for lipid classes identified was according to Pauling et al. [107].

### 4.9. Western Blot Analysis

Total protein was extracted from the organic phase of TRI Reagent^®^ (Sigma-Aldrich, Merck, Cat. No. T9424) treated tissue samples (5 mg DW), following the manufacturer’s protocol with modifications to ensure protein purity. Precipitated protein was washed with 0.1 M ammonium acetate in methanol and acetone, air-dried, and resuspended in 2% SDS. Protein concentration was quantified using the Micro BCA™ Protein Assay Kit (Thermo Fisher Scientific, MA, USA) against a BSA standard curve.

For SDS-PAGE, 40 µg of total protein per sample was denatured at 95 °C for 5 min in Lämmli sample buffer and resolved on 15% resolving gels with a 6% stacking gel. Proteins were transferred overnight to PVDF membranes (0.22 µm pore size; Durapore^®^, St. Paul, MN, USA) at 40 V and 4 °C in transfer buffer following Towbin et al. [108]. Membranes were briefly stained with Ponceau S to confirm equal protein loading.

Membranes were blocked in 5% skim milk in TBS-T (0.2% Tween-20) and incubated overnight at 4 °C with a rabbit polyclonal anti-oleosin antibody (OLE1, AT4G25140; PhytoAB, #PHY0954S) at 1:1000. The expected band size is ca. 19 kDa, and cross-reactivity with *E. nindensis* homologues was confirmed by the manufacturer. After washing, membranes were incubated with HRP-conjugated goat anti-rabbit IgG (Sigma-Aldrich) at 1:2000 for 2 h at room temperature. Signal detection was performed using WesternBright ECL (Advansta, CA, USA) and imaged with a ChemiDoc system (Bio-Rad, CA, USA). Three biological replicates, each with three technical replicates, were used.

## Figures and Tables

**Figure 1 plants-14-03360-f001:**
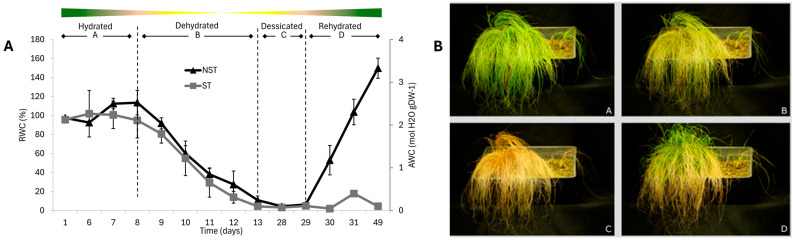
(**A**) Changes in relative water content (RWC) and absolute water content (AWC) over time in the resurrection plant *Eragrostis nindensis*, comparing desiccation-tolerant non-senescent tissue (NST) and desiccation-sensitive senescent tissue (ST) during drying and rehydration. Four physiological stages are indicated: A—Hydrated (100% RWC), B—Dehydrated (60–25% RWC), C—Desiccated (air-dry), and D—Rehydrated (12 h post-watering). The colour gradient illustrates chlorophyll loss during dehydration and its recovery upon rehydration. Breaks in the x-axis (days 1–6 and 13–28) are included to improve readability. Vertical bars represent standard errors (*n* = 6 plants, 3 technical replicates per plant). (**B**) Representative photographs of the four treatment stages corresponding to panel (**A**). Two-way ANOVA showed significant effects of time (*p* < 0.0001), tissue (*p* < 0.0001), and their interaction (*p* < 0.0001).

**Figure 2 plants-14-03360-f002:**
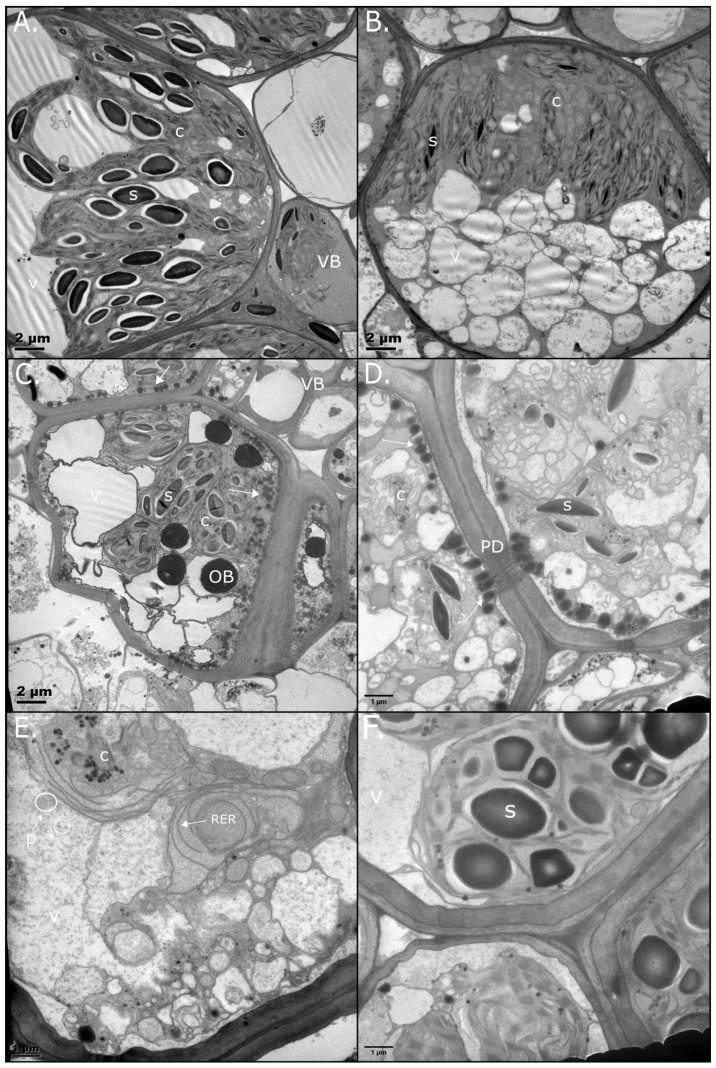
Transmission electron micrographs (TEMs) of bundle sheath (BS) cells from NST of *E. nindensis* during progressive drying (RWC) and rehydration (12 h). (**A**) At full hydration (100% RWC), chloroplasts (c) were structurally intact, actively photosynthesising, and accumulating starch (s). A vascular bundle (VB) is evident, affirming that this is a BS cell. (**B**) At 60% RWC, vacuoles (v) became fragmented, and low-density lipid droplets (LDs) appeared near the plasmalemma. (**C**) At 40% RWC, BS cell size was reduced but shape maintained; large osmophilic bodies (OBs) emerged and small lipid droplets accumulated along the cell wall (arrow). (**D**) At 25% RWC, thylakoids were dismantled, but plasmodesmata (PD) remained intact, indicating preserved cell-to-cell communication. LD accumulation was prominent (arrow). OBs were not as evident. (**E**) In the desiccated state (<10% RWC), extensive ribosomal endoplasmic reticulum (RER; arrowed) was observed, implying active protein synthesis. Darkly stained clusters resembling polyribosomes (p, circles) were observed. (**F**) At 12 h post-rehydration, chloroplasts were largely reassembled, and starch (s) granules were again present.

**Figure 3 plants-14-03360-f003:**
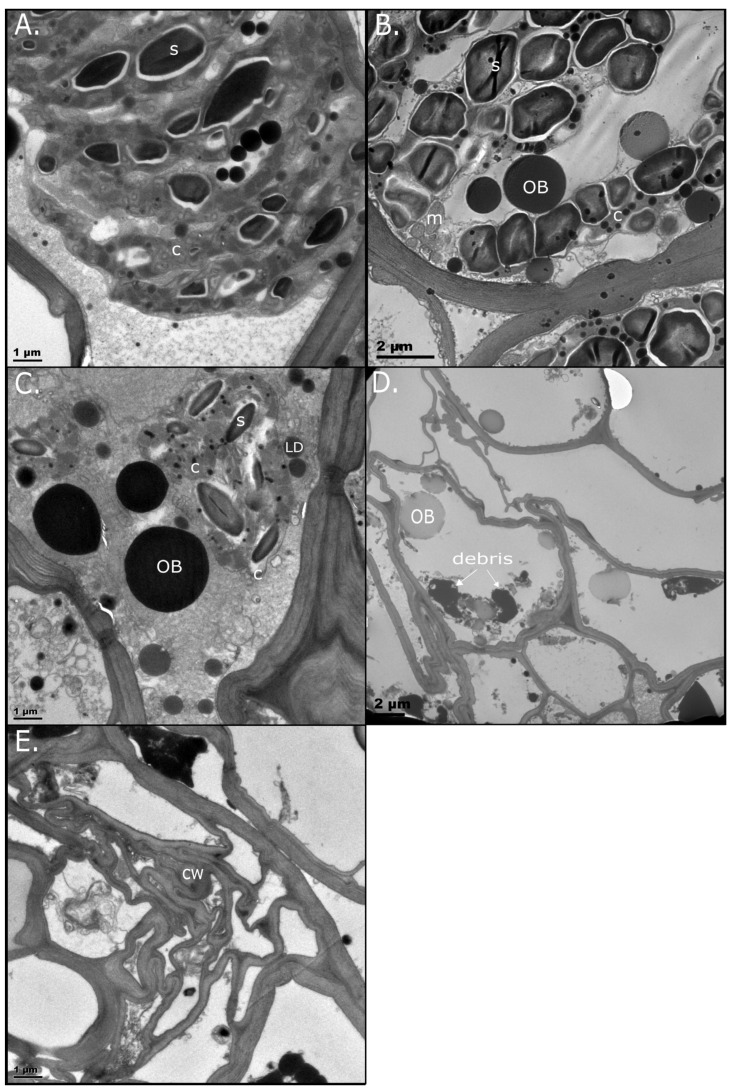
TEM of BS cells from ST of *Eragrostis nindensis* during drying (RWC, %) and after 12 h of rehydration. (**A**) At full hydration (100% RWC), chloroplasts (c) were intact and actively photosynthesising, with visible starch granules (s). (**B**) At 60% RWC, chloroplasts became indistinct, while large starch grains and OBs accumulated. Mitochondria (m) appeared structurally active. (**C**) At 25% RWC, chloroplasts were dismantled, starch content was reduced but still detectable, and large OBs were present alongside dispersed and small LDs. (**D**) In the desiccated state (<10% RWC), cell walls (cw) collapsed, and large OBs and cellular debris were evident. (**E**) After 12 h of rehydration, cells showed no structural recovery; cell walls remained collapsed.

**Figure 4 plants-14-03360-f004:**
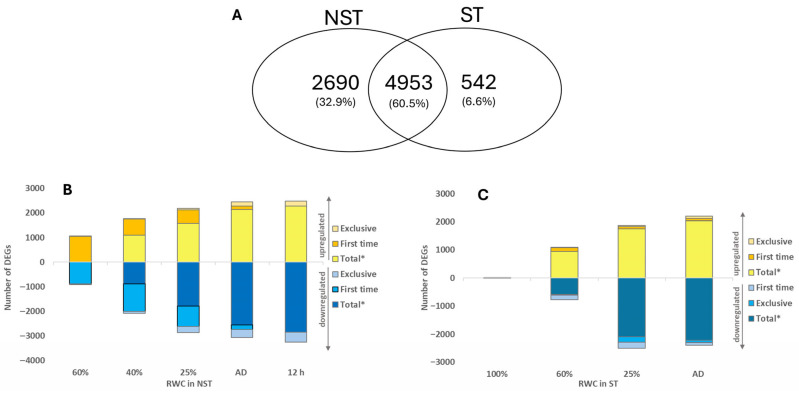
(**A**) Venn diagram presenting the proportion and number of differentially expressed genes (DEGs) commonly expressed across all samples, and exclusive genes per tissue type (NST, ST) during drying and rehydration. (**B**,**C**) Number of DEGs either accumulating (**B**) or diminishing (**C**) in NST or ST. Total* excludes genes that are expressed exclusively or for the first time. First time genes do not include exclusive genes.

**Figure 5 plants-14-03360-f005:**
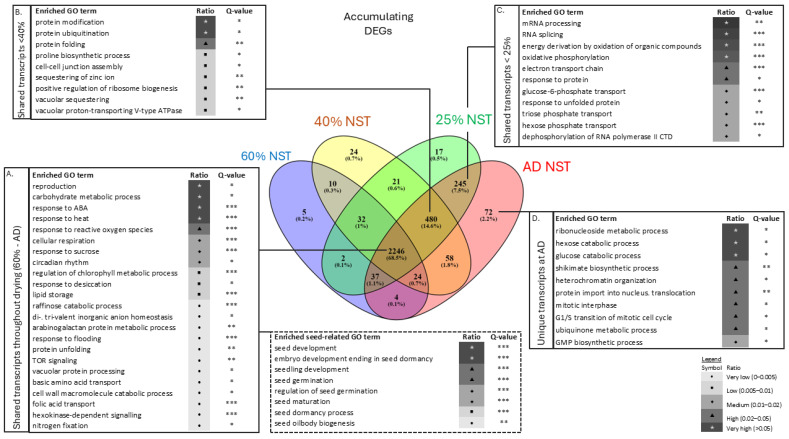
Drying-induced accumulating DEGs, represented by enriched Gene Ontology (GO) categories (*p* < 0.05), show distinct metabolic shifts in NST of the resurrection plant *Eragrostis nindensis* and display core traits of desiccation tolerance. The number of DEGs is relative to fully hydrated leaves (100% RWC, NST). GO categories of genes expressed throughout drying at different RWCs (shared genes) or only expressed in the desiccated state (air-dry (AD), unique genes) are depicted in boxes. The dotted box shows the seed-associated enriched GO categories that were expressed throughout drying. Significance was determined using BiNGO with FDR correction (q-value < 0.05; * q < 0.05, ** q < 0.01, *** q < 0.001). Ratio (of expected to observed number of genes) represents the degree to which each category is overrepresented as described in the legend.

**Figure 6 plants-14-03360-f006:**
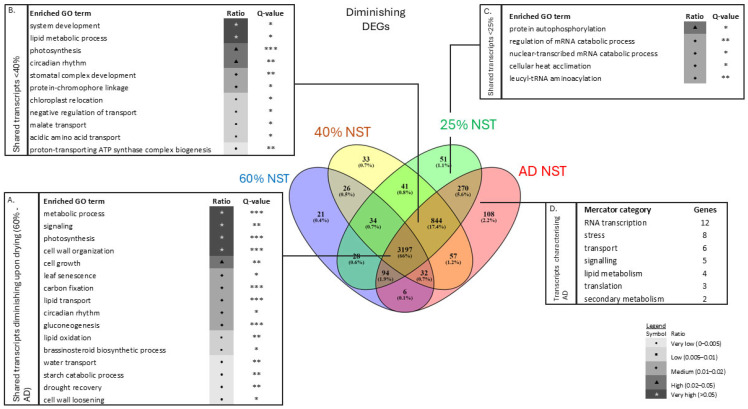
Drying-induced downregulated DEGs, represented by enriched GO categories (*p* < 0.05), show distinct metabolic shifts in NST of *Eragrostis nindensis*. The number of DEGs are relative to fully hydrated leaves (100% RWC, NST). Boxes summarise the most enriched GO categories of shared genes expressed during each drying stage. Mercator categories are shown for air-dry samples (AD, RWCs of <10%). Significance was determined using BiNGO with FDR correction (q-value < 0.05; * q < 0.05, ** q < 0.01, *** q < 0.001). Ratio (of expected to observed number of genes) represents the degree to which each category is overrepresented as described in the legend.

**Figure 7 plants-14-03360-f007:**
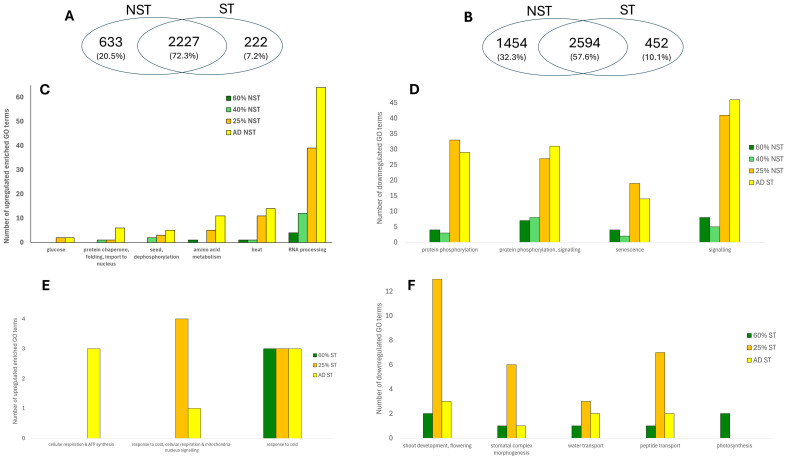
Differential gene expression and enriched GO categories during drying in *E. nindensis.* (**A**,**B**) Venn diagrams show shared and unique upregulated (**A**) and downregulated (**B**) DEGs between NST and ST during drying. (**C**,**E**) Enriched GO terms associated with upregulated DEGs in NST (**C**) and ST (**E**). (**D**,**F**) Enriched GO terms associated with downregulated DEGs in NST (**D**), and ST (**F**). Differential expression was defined as log_2_ fold change > 2 and FDR < 0.05 relative to the hydrated NST control (100% RWC).

**Figure 8 plants-14-03360-f008:**
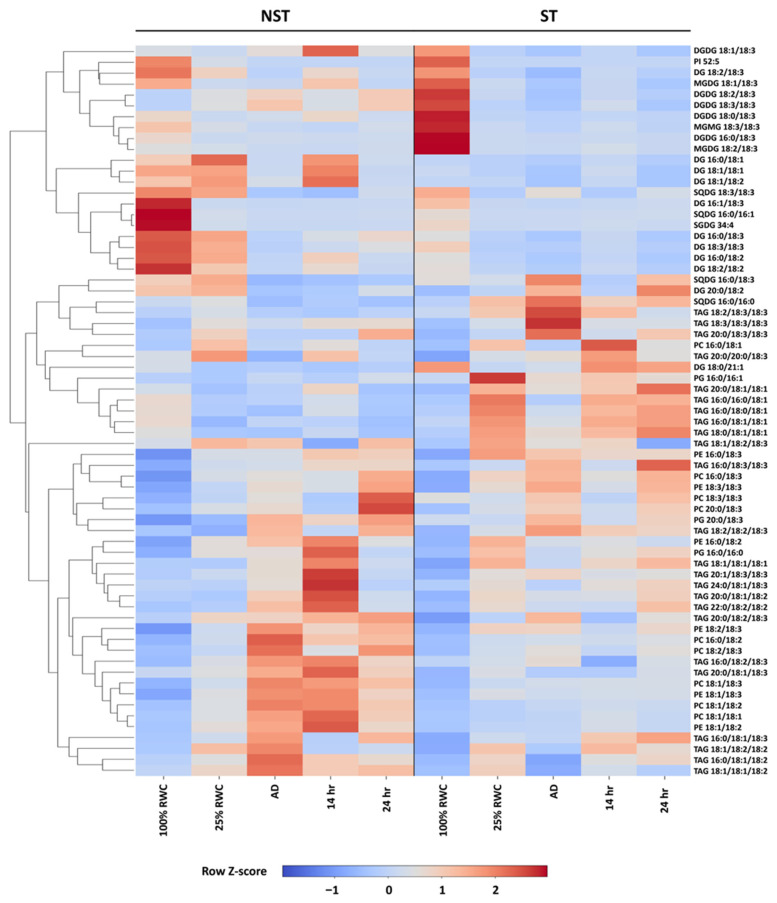
Heatmap with hierarchical clustering of lipid species during dehydration and rehydration. Metabolite abundances were normalised by row-wise Z-score, and hierarchical clustering was applied to group metabolites with similar response patterns. Columns represent experimental conditions in a fixed order. Warmer colours (red) indicate higher relative abundance compared to the metabolite’s mean, while cooler colours (blue) indicate lower abundance. A separate horizontal colour bar indicates the Z-score scale.

**Figure 9 plants-14-03360-f009:**
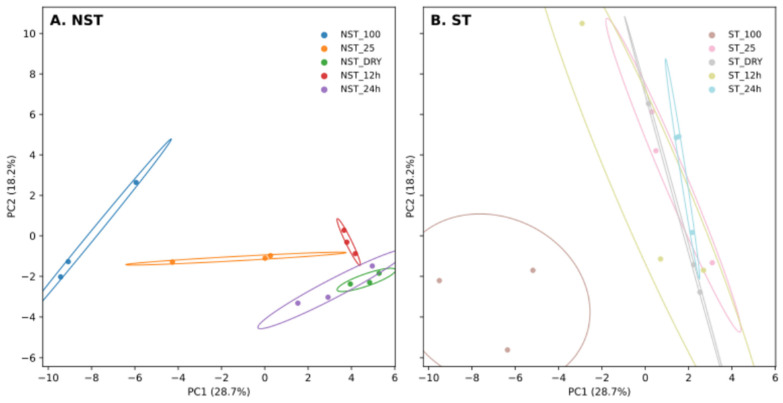
Principal Component Analysis (PCA) of lipid species during dehydration and rehydration. Scores for PC1 vs. PC2 are shown, with each point representing an individual sample. Panels display the same PCA projection computed once on the full dataset but visualised separately for (**A**) NST samples and (**B**) ST samples to improve legibility; positions are directly comparable across panels. Ellipses depict two-standard-deviation dispersion for each condition (bivariate SD ellipse; ca. 86% coverage under a Gaussian assumption). Axis labels report the percentage of variance explained by each principal component.

**Figure 10 plants-14-03360-f010:**
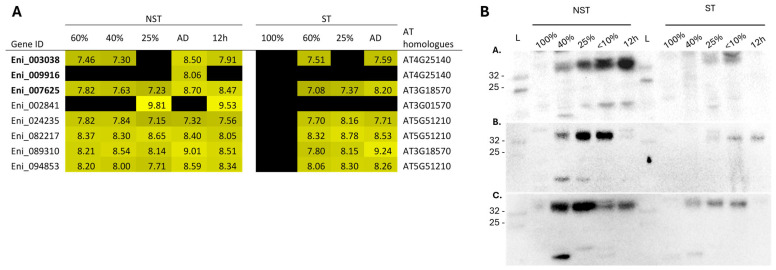
(**A**) Heatmap of eight differentially expressed oleosin transcripts and their corresponding *Arabidopsis thaliana* homologues identified in NST and ST during desiccation and rehydration. Differential expression is shown as log_2_-fold change relative to 100% NST, with lighter shades of yellow indicating higher fold change and black representing no change. (**B**) Western blot analysis of oleosin protein expression using the OLE1 antibody with three biological replicates (A.–C.). A dominant band at ca. 33 kDa was consistently observed in NST during drying, with the strongest expression at lower RWCs (25% and 10%). In contrast, ST showed faint or undetectable signals at the same position. Molecular weight was estimated using the Colour Prestained Protein Standard, Broad Range ladder (L; 11–245 kDa, New England Biolabs). The observed molecular weight shift from the expected 19 kDa may be due to mono-ubiquitination or interaction with oleosin-like proteins such as caleosins [45].

**Table 1 plants-14-03360-t001:** Enriched GO terms (*p* < 0.05) up- or downregulated during rehydration (12 h post-watering) in NST of *Eragrostis nindensis* indicate quick resumption of metabolism. Ratio (of expected to observed number of genes) represents the degree to which each category is overrepresented.

Expression Trend	GO Term	Description	Ratio
Accumulating DEGs	GO:0017003	protein–heme linkage	0.0136
	GO:0071258	cellular response to gravity	0.0136
	GO:0006089	lactate metabolic process	0.0204
	GO:0009438	methylglyoxal metabolic process	0.0204
	GO:0018315	molybdenum incorporation into molybdenum–molybdopterin complex	0.0136
	GO:0042040	metal incorporation into metallo–molybdopterin complex	0.0136
	GO:0046185	aldehyde catabolic process	0.0204
	GO:0019243	methylglyoxal catabolic process to D-lactate via S-lactoyl-glutathione	0.0204
Diminishing DEGs	GO:0030245	cellulose catabolic process	0.0135
	GO:0035445	borate transmembrane transport	0.0101

## Data Availability

The original RNA-Seq data presented in the study are openly available in the Short Reads Archive at http://www.ncbi.nlm.nih.gov/bioproject/1310038 (submitted on 16 September 2025).

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
