# Peer review of "Defying Death: A Multi-Omics Approach to Understanding Desiccation Tolerance and Senescence in *Eragrostis nindensis"

_plants, 2025, doi:10.3390/plants14213360_

Round 1

Reviewer 1 Report

Comments and Suggestions for Authors

The present article complies with the journal´s aims and scope and it provides interesting results to the public. Results and discussions are well presented. However attention must be payed to details such as red lines that were found in some sections perhaps a result of previous revisions. Also, the quality of all images must be improved they seem to be pixeled and difficult to follow. For instance in figure 5 seems to be a bit crowded with information. The colors chosen for the description of the variance of ratio vs q value are not easy to correlate with the results found within the DEG gene groups. Same in figure 6. In figure 7 lettering is way to small and difficult to read. Figure 8 in the heatmap also letters are too small to read at first sight. And in figure 10 the table and image provided seem a bit pixeled. Please improve the quality. 

Reviewer 2 Report

Comments and Suggestions for Authors
  • The Introduction defines desiccation tolerance as surviving <30% RWC, yet E. nindensis is described as surviving >90% water loss. Please reconcile these thresholds to avoid confusion (e.g. explicitly state this means ~10% RWC).

  • The text attributes ST senescence to age but does not explain how leaf age was standardised or confirmed. Include explicit criteria (e.g. chlorophyll index, developmental stage) to show ST are indeed older and age-synchronized.

  • The claim that NST but not ST recovered after rehydration is visually shown but not statistically tested. Please include survival or viability quantification (e.g. % leaf recovery, statistical test used).

  • The observation that LDs cluster near plasmodesmata as “potentially preserving cell-to-cell communication” is speculative. Provide supporting references or temper the language.

  • (Lines 586–587) The Methods state ST at 40% RWC and 12 h rehydration were not sampled. This creates gaps in the comparative analysis. Discuss how this affects interpretation of differential responses between NST and ST.

  • (Lines 615–617) Because ST was sequenced only at 100%, 60%, 25% and AD, but NST at more stages, it is unclear whether observed gene expression differences reflect true biology or sampling bias. Please clarify this limitation and its impact on downstream analyses.

  • Protein validation and molecular weight discrepancy (Lines 381–387) The immunoblot shows bands at 33 kDa vs. the expected 19 kDa. While possible explanations are listed, include molecular weight markers on the blot figure and, if possible, confirm identity via mass spectrometry or peptide mapping.

  • (Lines 243–258) The text states metabolic recovery occurs during rehydration, yet only NST was analysed. Explicitly state that no transcriptomic or lipidomic data exist for ST rehydration and temper claims about their comparative recovery.

Comments on the Quality of English Language

The manuscript is generally well written, but some sentences are long and densely packed, especially in the Discussion (e.g. Lines 520–539). Breaking long sentences into shorter, clearer statements will enhance readability. A thorough English proofreading by a native speaker is recommended to correct minor grammar and syntax issues.

Reviewer 3 Report

Comments and Suggestions for Authors

Dear Authors,

I carefully read your manuscript and would like to share my feedback for improving its quality as follows:

Abstract:

  1. Add 1–2 sentences upfront highlighting why Eragrostis nindensis is a valuable model for studying desiccation tolerance, and what gap in knowledge this study addresses.
  2. Briefly indicate the approach or scope (e.g., RNA-seq, TEM, lipid profiling by LC-MS) without going into full detail, to improve scientific rigor and transparency.
  3. Clarify how polyunsaturated TAGs and lipid droplets specifically contribute to membrane stabilization and recovery.

Introduction:

  1. Explicitly state why existing knowledge on desiccation tolerance and lipid metabolism is incomplete, and what specific gap this study addresses.
  2. Condense some general background (e.g., the section on osmoprotectants and antioxidants, lines 37–41) and shift more quickly to NST vs ST comparisons and lipid metabolism.
  3. More directly connect prior lipid findings in other resurrection plants to the gaps in understanding for E. nindensis.
  4. Lines 74 – 83: Frame the study as testing a clear hypothesis.

Results:

  1. Some results sections (e.g., 2.1 and 2.2.2) contain too much descriptive detail that reads like a methods narrative rather than a concise presentation of findings. Consider condensing overly detailed morphological descriptions and focusing on key patterns.
  2. The text mentions enriched GO terms and DEG counts but does not consistently provide p-values, FDR thresholds, or fold-change cutoffs in the main narrative (only in figure legends or supplement references). Explicitly stating these thresholds in the results would strengthen reproducibility.
  3. For physiological measurements (RWC, AWC), standard errors are shown in figures, but no statistical comparisons are reported in the text (e.g., ANOVA results, significance between NST and ST).
  4. Abbreviations: Make sure definitions are introduced once and consistently used thereafter.

Discussion:

  1. Several sections (e.g., 3.2, 3.5) would benefit from comparisons with other resurrection plants or stress models to highlight novelty
  2. In the discussion please focus on hypothesis, and explain the novelty.
  3. trengthen integration across datasets.
  4. Provide deeper mechanistic insights rather than descriptions.
  5. Avoid redundancy (esp. metabolomics).
  6. Reframe speculative statements more cautiously.
  7. Explicitly state why ST fails compared to NST.
  8. Add broader significance and future directions for plant stress biology/crop improvement.
  9. Improve clarity by breaking down overly dense sentences and defining jargon.

Materials and methods:

  1. Seed source: Specify if permits/permissions were obtained for seed collection (important for biodiversity regulations).
  2. Nutrient preparation: Clarify exact concentrations (e.g., “2% calcium” → specify whether calcium chloride, calcium nitrate, etc.).
  3. Fertilizer: Phostrogen composition (NPK values) should be provided for reproducibility.
  4. Randomization: Mention whether trays were randomized during growth to minimize positional/light effects.
  5. Clarify environmental conditions (temperature, humidity, airflow) during drying—critical for reproducibility.
  6. Sampling limitations: Explain why ST samples at 40% and 12 h were excluded, and discuss potential bias this introduces.
  7. Provide statistical justification for “minimum of three biological replicates”—this is low for omics studies; reviewers may flag power issues.
  8. State whether technical replicates (RNA-seq, lipidomics, Western blots) were performed.

Conclusion:

I couldn’t find a conclusion section in this manuscript. Please add it in the revised version.

Reviewer 4 Report

Comments and Suggestions for Authors

Dear authors,

The title of the article is very attractive, and the manuscript is clearly written. The authors present their data and results logically and coherently, making the manuscript easy to follow. Figures are well-designed, properly labeled, and relevant to the study’s objectives. Raw data are reported and appear to be provided in accordance.

The experimental design is appropriate for the study’s objective, and the approach is relevant and well-structured.

The findings align with the research objectives and are supported by the raw data provided. The results are robust, and the authors’ interpretations are logical and consistent with the evidence presented.

I have only some remarks for the revised version before publication:

  1. Fig. 10B showed the Western blot of OLE1 protein in E. nindensis with its expected molecular weight (M.W) of 19 kDa. However, the actual detected size was around 33 kDa. Even though authors have been given the discussion about this observation (L381-387), it will be better if authors include a positive control (using Arabidopsis OLE1 protein, which is based on the antibody of E. nindensis OLE1 was generated. Loading the Arabidopsis OLE1 protein (positive control) in parallel with the protein of E. nindensis in SDS-PAGE gel allows easier tracking of the Western blot data and comparison, since it shows a big difference in the M.W range of E. nindensis OLE1.
  2. Combine 2 paragraphs/sections: “Lipid metabolism…resurrection plants” and “Triacylglycerol (TAG)… storage organelles” into 1 section.
  3. The authors used TRIzol (Sigma-Aldrich) (L614) and TRIzol® (L692) for RNA isolation and total protein extraction, respectively. Are they the same one? Please include the catalog number, product code of these TRIzol chemicals in the main text of the manuscript.
  4. Fig.8 is presented in low quality. Please replace with a high-resolution one.
  5. What does "ca." (L36, 88, 381, 383, 705) mean?
  6. Italicize the scientific name of the plant: E. nindensis (L93)

With regards,

Round 2

Reviewer 3 Report

Comments and Suggestions for Authors

I guess the author(s) don’t agree to my suggestions. I cannot accept it in its current form. 
I will let the editor decide. Thank you. 

Kind Regards,

Author Response

Dear Reviewer

One editor has agreed that our paper be published as is, having been reviewed by 3 other reviewers and having attended to all reasonable comments from reviewers. The second requested that we at least attend to your request to mention in text, in addition to the 4 other times our paper mentions this, the following sentence.  

Genes were classified as differentially expressed when log₂ fold change exceeded ±2.0, with a false discovery rate (FDR; q ≤ 0.05, [95]) applied relative to the control (100% RWC, NST)”. 

I trust this is sufficient to enable publication in Plants

regards

Jill M. Farrant